# New Opportunities in Heart Failure with Preserved Ejection Fraction: From Bench to Bedside… and Back

**DOI:** 10.3390/biomedicines11010070

**Published:** 2022-12-27

**Authors:** Alfredo Parra-Lucares, Esteban Romero-Hernández, Eduardo Villa, Sebastián Weitz-Muñoz, Geovana Vizcarra, Martín Reyes, Diego Vergara, Sergio Bustamante, Marcelo Llancaqueo, Luis Toro

**Affiliations:** 1Critical Care Unit, Department of Medicine, Hospital Clínico Universidad de Chile, Santiago 8380420, Chile; 2MD PhD Program, Faculty of Medicine, Universidad de Chile, Santiago 8380420, Chile; 3Division of Internal Medicine, Department of Medicine, Hospital Clínico Universidad de Chile, Santiago 8380420, Chile; 4School of Medicine, Faculty of Medicine, Universidad de Chile, Santiago 8380420, Chile; 5Coronary Care Unit, Cardiovascular Department, Hospital Clínico Universidad de Chile, Santiago 8380420, Chile; 6Division of Nephrology, Department of Medicine, Hospital Clínico Universidad de Chile, Santiago 8380420, Chile; 7Centro de Investigación Clínica Avanzada, Hospital Clínico, Universidad de Chile, Santiago 8380420, Chile

**Keywords:** heart failure, preserved ejection fraction, cardiology, therapeutics, translational medicine

## Abstract

Heart failure with preserved ejection fraction (HFpEF) is a growing public health problem in nearly 50% of patients with heart failure. Therefore, research on new strategies for its diagnosis and management has become imperative in recent years. Few drugs have successfully improved clinical outcomes in this population. Therefore, numerous attempts are being made to find new pharmacological interventions that target the main mechanisms responsible for this disease. In recent years, pathological mechanisms such as cardiac fibrosis and inflammation, alterations in calcium handling, NO pathway disturbance, and neurohumoral or mechanic impairment have been evaluated as new pharmacological targets showing promising results in preliminary studies. This review aims to analyze the new strategies and mechanical devices, along with their initial results in pre-clinical and different phases of ongoing clinical trials for HFpEF patients. Understanding new mechanisms to generate interventions will allow us to create methods to prevent the adverse outcomes of this silent pandemic.

## 1. Introduction

Heart failure (HF) with preserved ejection fraction (HFpEF) is a clinical syndrome that is currently subject to intense debate regarding the pathophysiological mechanisms underlying its generation and progression [1,2]. These mechanisms are crucial to finding therapeutic options that can modify the natural history of the disease since approximately 50% of new diagnoses of heart failure correspond to patients with non-reduced ejection fraction (EF) [3,4]. Likewise, morbidity and mortality in this patient group are comparable to patients with reduced ejection fraction (HFrEF) [5,6]. Although cardiovascular events cause most mortal events in HFpEF, a great proportion is due to comorbidities in this syndrome, unlike HFrEF [7].

At present, few pharmacological therapeutics for the management of HFpEF patients have proven to reduce morbidity and mortality. All of them have been extensively reviewed. This review aims to present a summary in diagnosis and management in HFpEF focused on new targets such cardiac fibrosis, alterations in calcium handling, NO pathway, and systemic inflammation, among others. Comprehending these interventions will enable understanding of the most promising methods to prevent adverse outcomes of this silent pandemic.

## 2. Definitions in HF

Heart failure’s (HF) definition has changed through time. The previous standard was based on the Framingham study [8], but current guidelines define it as a clinical syndrome secondary to a structural and/or functional abnormality of the heart that results in elevated intracardiac pressures (i.e., an impaired ventricular filling) and/or inadequate cardiac output [9].

HF can be classified according to its etiology, stage, functional capacity, trajectory, and left ventricular ejection fraction (LVEF). The latter being of special significance, because of differing prognosis and response to treatments. Moreover, most clinical trials select patients based on LVEF [10].

Reduced LVEF (HFrEF) is defined as ≤40%, while if LVEF is between 41% and 49%, it is defined as HF with mildly reduced LVEF (HFmrEF). This “new” classification was suggested considering that both groups of patients may benefit from similar therapies. In a third category, patients with LVEF ≥ 50% who have a clinical presentation plus objective evidence of cardiac structural and/or functional abnormalities have HF with preserved LVEF (HFpEF) [9,10].

## 3. Clinical Presentation and Diagnosis

HF is suspected in a patient with cardiovascular risk factors (CVRF). Hypertension and obesity are highly present in patients with HFpEF [11]. Other CVRF include diabetes mellitus, chronic kidney disease, alcohol consumption, and a family history of cardiomyopathy or sudden death, as well as specific etiologies that can directly impair cardiac structure and/or function such as a history of a congenital heart disease, a previous myocardial infarction (MI), cardiotoxic chemotherapy, or valve disease [9]. 

If symptomatic, patients generally present with signs and symptoms of clinical congestion, that result from elevated cardiac filling pressures. Typical HF symptoms include dyspnea, which can be classified according to the New York Heart Association (NYHA) functional classification, and reduced exercise tolerance, which is a canonical symptom in HFpEF [12]. Other symptoms that may be present are orthopnea, paroxysmal nocturnal dyspnea, fatigue, tiredness, and leg edema, while specific signs include the presence of jugular vein distention, hepatojugular reflux, third heart sound, and laterally displaced apical impulse. Nonetheless, they lack sufficient accuracy to be used alone to make the diagnosis of HF [9]. 

For an initial assessment, the European Society of Cardiology and the American Heart Association recommend an electrocardiogram (ECG), which will seldom be normal, often presenting abnormalities such as atrial fibrillation (AF), Q waves, LV hypertrophy, and a widened QRS complex [10]. Furthermore, an initial laboratory evaluation should include a complete blood count, urinalysis, serum electrolytes, blood urea nitrogen, serum creatinine, glucose, fasting lipid profile, liver function tests, iron studies, and thyroid-stimulating hormone levels [9]. A chest X-ray is also recommended to investigate differential diagnosis, and because it can show findings that support the presence of HF [10].

For diagnostic confirmation an echocardiography is mandatory, since it may determine LVEF, as well as providing information about heart structure and function [10]. Most relevant measurements that may be present in HFpEF are an elevated E/e’ ratio (early mitral inflow velocity and mitral annular early diastolic velocity) and an increase in the pulmonary artery systolic pressure [13]. 

Assessment of natriuretic peptides (NPs) is recommended, if available, considering that low levels may help exclude the presence of HF because of their high negative predictive values, while higher levels have a high positive predictive value for HF [10], although this must be interpreted with caution since patients with HFpEF have NP levels that are lower than those of patients with HFrEF, and may even be normal especially considering the high prevalence of obesity in this group of patients [11], moreover, there are other sources for elevated NPs such as renal or liver dysfunction [9,10].

Integrated diagnostic approaches have recently gained increasing importance. These are the H_2_FPEF [13] and the HFA-PEFF [14] scores. Both tools assign patients to a group with low, intermediate, or high likelihood of HFpEF, allowing further diagnostic methods only for those with an intermediate score. The former uses a composite evidence-based approach, while the latter involves a systematic step-by-step algorithm based on an expert consensus. They have shown, independently, a good discriminatory capacity [15,16], but discordant results when applied to the general population [17]. 

Finally, screening for etiologies is the last step, looking for cardiovascular and non-cardiovascular comorbidities that impair myocardium function, for example, ischemic disease, toxic (medications, radiation, substance abuse, heavy metals), infiltrative, immune, or inflammatory pathology. Or alterations in loading conditions, i.e., hypertension, acquired or congenital valvular disease, volume overload (renal failure), and cardiac rhythm disturbances, among others [14]. 

In conclusion, although numerous diagnostic approaches and tools have been proposed, HFpEF diagnosis still needs to be improved for clinicians and researchers, and a unifying criterion is still required.

## 4. Non-Invasive Complementary Evaluation

As initial assessment of HFpEF patients is mandatory to find objective evidence of structural, functional, and serological abnormalities. A summary of the main features in transthoracic echocardiographic is presented in Table 1.

Serum biomarkers are key points in the study of heart pathologies and Table 2 shows the different serology test used in this group of patients.

In addition, the gold standard to evaluate morphologic changes in HFpEF patients is cardiac resonance magnetic imaging. This complementary study is useful for identifying etiologies associated with imaging characteristics. Table 3 summarizes the principal findings according to radiologic findings.

## 5. Hemodynamic Abnormalities

Hemodynamic abnormalities can be assessed in HFpEF to demonstrate functional changes [14]. The hallmarks amongst cardiac factors for hemodynamics changes are diastolic dysfunction, subtle systolic dysfunction, pulmonary hypertension, and right ventricular dysfunction. In healthy adults and early stages of HFpEF left ventricular end diastolic pressure (LVEDP), right atrial pressure (RAP), mean pulmonary artery pressure (PAP), and pulmonary capillary wedge pressure (PCWP) cannot have changes and some of these are unmasked with an exercise-tolerance test. This contrasts with advanced HFpEF which shows an increase in all measurements [31]. 

A gold standard approach for the measurement of these features is left heart catheterization, but it is technically difficult in a typical health center. Therefore, right heart catheterization could be a surrogate for this aim and should be considered in the work up of these patients. The definitive measure for HFpEF diagnostics is mean PCWP (mPCWP) at rest ≥ 15 mmHg in the presence of normal LV-end diastolic volume index. The latter reveal a reduced LV distensibility. However, a normal mPCWP at rest could be elevated in exercise or show decompensated conditions in a compensated HFpEF patient [14]. Only hemodynamic measurements in structured exercise conditions may correct classify intermediate score patients [32,33]. Furthermore, exercise-based approaches can determine a prognostic in these patients [14]. The main cut-off values for rest and exercise for right heart catheterization are presented in Table 4.

## 6. General Pathophysiology in HFpEF

In the past, the main feature associated with HFpEF was diastolic dysfunction. For this reason, both terms were used as synonyms twenty years ago [35]. Nevertheless, we find diastolic dysfunction in preserved and EF is reduced. Non-cardiac abnormalities appear in this manner, from the molecular to the systemic level, as presented in Figure 1. Hence, the advance in molecular mechanisms in basic cardiology research and new concepts enable a greater understanding of more targets in this syndrome.

At the molecular level, chronic inflammation is a leading cause of HFpEF due to dysregulation in the humoral immune system. High levels of cytokines such as interleukine-1, -6 (IL-1, IL-6) and tumor necrosis factor-α (TNF-α) in blood samples, cell adhesion molecules, and other mediators can infiltrate the myocardium, leading to chronic damage [36]. In the same way, cardiac fibrosis plays a central role in fibroblast replacement in the heart due to activation gene expression of myofibroblast through some profibrotic factors (TGFb, IL-11, AngII) [37]. Another well-known pathway is nitric oxide (NO), which is becoming a potential new target because cGMP levels are decreased in HFpEF related to oxidative stress, endothelial dysfunction, and inflammation. NO is a promissory treatment since it corrects cardiac fibrosis, immune dysregulation, and gap junction remodeling [38]. At this level, reactive oxygen species (ROS) lead to mitochondrial damage [39], with insufficient ATP production activating downstream pathways associated with cardiac fibrosis, inflammation, and diastolic dysfunction. Lastly, calcium is broadly studied due to calcium handling in cardiomyocytes being finely regulated related to the calcium-induced calcium release mechanism, best known as the calcium transient that drives muscle contraction. However, diastolic dysfunction could appear to be comorbidity-dependent in HFpEF [40].

At the cellular level, endothelial progenitor cells (EPCs) are necessary for endothelial repair, and EPCs can be recognized by their CD34+ receptor on the surface. HFpEF patients have lower and impaired circulating levels of CD34+ cells [41]. The reduced EPCs are related to NO imbalance, elevated ROS production, and a systemic proinflammatory state [42]. 

Finally, at the systemic level, neurohumoral and mechanical properties of cardiac function are impaired. Endothelin and adrenomedullin pathways are activated in this syndrome associated with pulmonary hypertension, resulting in worse right ventricular diastolic reserve and oxygen consumption dysregulation in a patient with reduced cardiac output and elevated left ventricular (LV) filling pressure [43]. 

These mechanisms will be exhaustively addressed at diverse levels, focusing on their preliminary results in pre-clinical and different-phase clinical trials.

## 7. The First Choice Drug: Diuretics

The ESC 2021 guideline has a low-grade recommendation in the use of a diuretic within HFpEF [44]. This is mainly based on a summary of studies from randomized controlled trials for diuretics in patients with congestive heart failure with no distinction of LV ejection fraction [45].

Even though acute decompensation is one of the most common causes of hospitalization in heart failure, congested patients have some significant differences in hemodynamic features [46]. A small study showed differences in volume overload across HFpEF. Measuring total blood volume with labeled albumin and red blood cell mass, revealed an increase excess of volume in HFrEF compared to HFpEF evidenced by less intravascular volume expansion but more interstitial fluid congestion [47].

In the same line, the diuretic response may differ between both diseases. Despite the wide use of diuretics, there is still a lack of a standardized protocol for depletive therapy and no differences have been reported between bolus or continuous infusion in acute heart failure management [48]. A recent study compared the high-dose with the low-dose diuretic strategy separately in HFrEF and HFpEF, revealing a higher net fluid loss and weight change in HFrEF at 72 h with the high-dose strategy, and with no differences in HFpEF on efficacy of intensive therapy. Interestingly, higher doses of diuretics were associated with impaired renal function in HFpEF, with a higher proportion compared with HFrEF [49]. Similarly, high doses of diuretics were associated with lower mortality rates in HFrEF but with no significant differences in HFpEF. 

Taken together, there is a need for more individualized therapy considering phenotypes of presentation at admission. Further research is needed to improve the accuracy of congestion in a multiparameter approach with clinical and laboratory findings to lead depletion therapy in HFpEF and HFrEF. 

## 8. Calcium Handling 

### 8.1. Physiology and Pathology Pathway 

Cardiomyocyte contraction is generated through calcium dynamics and has become a potential target for treatment in vivo and in silico [50]. Calcium-induced calcium release finishes with actin–myosin coupling. This process is known as the calcium transient and is regulated by several proteins to maintain basal intracellular calcium levels. This transient starts with an action potential trigger that activates L-type calcium channels (LTCC) that open ryanodine receptors in the sarcoplasmic reticulum (the main reservoir of calcium), releasing more calcium, this activates a large proportion of LTCC in the plasma membrane of cardiomyocytes with a strong rise in intracellular concentration. After actin–myosin activation, cardiac contractile cells must diminish their high calcium levels due to its toxicity if it becomes chronic. For the return to basal concentration, there are several proteins, for example, the SERCA protein, that return calcium to the sarcoplasmic reticulum or sodium–calcium exchanger (NCX) that enable the release from the intra- to extracellular space [51,52]. Furthermore, transient receptor potential (TRP) family channels participate in calcium homeostasis by augmenting calcium entry to cells and associating with hypertrophic remodeling [53]. 

In animal models, there are differences in the calcium handling between heart failure with preserved and reduced EF. In the first case, there is impaired relaxation due to the spatiotemporal inhomogeneity of the calcium decay in the calcium transient throughout the myocyte and alteration in the phosphorylation of some proteins involved, for example, ryanodine [54,55,56]. Furthermore, irregular calcium transients during the heartbeat have been observed [57]. 

Despite these observations, it is controversial if calcium dynamics are affected primarily or if it is related to the comorbidities that are present in HFpEF patients, such as hypertension, diabetes, obesity, or other systemic proinflammatory conditions [40,55,58,59].

### 8.2. Calcium Dynamics Approach Therapies

#### 8.2.1. Calcium Channels Blockers

Nifedipine inhibits calcium entry and reduces calcium overload. For this purpose, the DEMAND study (NCT01157481) was designed in which participants receive Nifedipine 10 to 60 mg once a day. Its primary outcome is a composite of mortality and hospitalization for HF. The target group has hypertension or coronary artery disease and HFpEF. Preliminary results show improvement in LV filling pressures, but as yet there are no complete results available.

#### 8.2.2. Ranolazine

An inhibitor of the late sodium current that prevents NCX calcium entry and overload was used in the RALI-DHF study [60] to improve diastolic calcium-associated relaxation parameters. However, it could only demonstrate diminished pulmonary capillary wedge pressure in 40 patients. The RAZE study (NCT01505179) tried to demonstrate better exercise capacity but only recruited ten from 40 patients, and there are no published results.

#### 8.2.3. B-Blockers

β-adrenergic signaling phosphorylates different calcium-handling proteins related to the calcium transient to augment or diminish their function. Nevibolol in the SENIORS study [61] with 2128 patients shows a reduction in mortality in HFrEF. On the other hand, carvedilol in the J-DHF trial could not improve outcomes in the subjects. Currently, we are waiting for results from a clinical trial comparing nevibolol and carvedilol in older adults (NCT02619526).

#### 8.2.4. Istaroxima

This Na/K ATPase inhibitor and SERCA activator promotes an increase in calcium transient amplitude and is being tested in HFpEF [62]. A clinical trial in early phase 1 is finishing his recruitment period to show an improvement in cardiac filling pressures (NCT02772068).

#### 8.2.5. Cardiac Glycosides

Digoxin is an old strategy for heart failure. It can increase the amplitude of the calcium transient. The DIG-PEF study [63] recruited patients with EF > 45%, sinus rhythm, angiotensin-converting enzyme inhibitors, and diuretics. Digoxin did not show a modification in mortality or cardiovascular hospitalizations.

## 9. Antifibrotic and Anti-Inflammatory Agents

### 9.1. Role of Inflammation

Cumulative studies have shown that inflammation is vital in the progression of myocardial damage, structural remodeling on the extracellular matrix (ECM), and adaptation of cardiomyocytes, leading to fibrosis [64] and diastolic dysfunction [65].

The first evidence of inflammation in HFpEF came from the observation that heart failure patients have higher levels of circulating inflammatory cytokines such as TNF-α [66,67], IL-1, IL-6, IL-8, and monocyte chemotactic protein (MCP)-1 [68]. Furthermore, several molecules have been described as inflammation biomarkers at higher levels in HFpEF including growth differentiation factor (GDF)-15, pentraxin-3 (PTX3), neopterin, and ST2. Inflammation has also been reported in endomyocardial biopsies of HFpEF, accounting for inflammatory cell recruitment in myocardial tissue [69].

GDF-15, a member of the transforming growth factor-beta cytokine superfamily, is a marker of cell injury and inflammation recently proposed as a biomarker with prognostic value [70,71,72]. PTX3, an acute phase response protein, is elevated in HFpEF as well as in subjects without heart failure symptoms but with LV diastolic dysfunction. Neopterin, an active molecule produced by activated macrophages acting as a pro-oxidant and promoting cell apoptosis, is higher in HFpEF patients [6]. ST2, a member of the IL-1 family of receptors with transmembrane ST2L and a soluble isoform (sST2) is easily measured in circulation, is elevated in patients with autoinflammatory conditions [73], and recently has been associated with the presence of diabetes mellitus, atrial fibrillation, renal dysfunction, and other comorbidities among patients with HFpEF [74]. Moreover, in an acute decompensated patient, the levels of PTX3, IL-6, TNF-α, and high sensitivity c-reactive protein (hsCRP) are higher compared to stable disease [75].

Evidence collected from an elevation of inflammation biomarkers raises some questions about the genesis of inflammation; whether in HFrEF cardiomyocyte damage is the crucial phenomenon triggering adaptative inflammation, fibrosis, and lately, systemic consequences. In HFpEF inflammation is proposed as a framework in which metabolic comorbidities, such as the presence of obesity, diabetes, anemia, hypertension, and others profoundly impact cardiac metabolism and inflammatory genesis [76,77]. 

The mechanistic link between metabolic conditions and inflammation needs to be fully understood. In one way, the transcriptional factor NFκB signaling pathway can be activated by metabolic stress leading to macrophage M1 polarization and secondary proinflammatory cytokines, chemokines, and adhesion molecule production, accounting for HFpEF obesity-related effects [78]. Although the role of macrophages in HFpEF is controversial [79], biopsies show that cardiac macrophages double in abundance compared to control [69]. One study showed that the culture of healthy donor monocytes with HFpEF patient-derived sera promoted M2 macrophage features, evidenced by altered morphology and genes and leading to a profibrotic phenotype [80]. Furthermore, evidence in mice revealed that either by hypertension or advanced age, cardiac macrophages expand with diastolic dysfunction leading to impaired myocardial relaxation and increased myocardial stiffness [81].

Particularly, TNF-α and IL-6 have pleiotropic effects over macrophages, fibroblasts, microvasculature, and cardiomyocytes [68], perturbing calcium homeostasis, triggering the apoptosis response, stimulating the synthesis of other cytokines, upregulating inducible nitric oxide synthase, degrading ECM and inducing endothelial adhesion molecules such as intercellular adhesion molecule (ICAM)-1 and vascular cell adhesion molecule (VCAM)-1 [82]. 

Attending to previously described aspects of HFpEF, some clinical efforts have identified inflammation and fibrosis as potential therapeutic targets.

### 9.2. Antifibrotic Agents

#### 9.2.1. Pirfenidone

The recently conducted PIROUETTE (Pirfenidone in Patients with Heart Failure and Preserved Left Ventricular Ejection Fraction) phase 2 trial with pirfenidone, an antifibrotic agent approved for idiopathic pulmonary fibrosis, showed some promising results [83]. This randomized, placebo-controlled trial included symptomatic patients with HFpEF within 12 months of diagnosis, defined by LVEF > 45%, and myocardial fibrosis assessed by cardiac magnetic resonance (CMR). However, this study showed promising results mainly because of the reduction in cardiac fibrosis in 1 year. However, the trial results should be interpreted carefully. CRM has gained relevance in fibrosis estimation due to extracellular volume (ECV) in HFpEF, and its cut-off points revealed a correlation with the lower composite outcomes of hospitalization and cardiovascular mortality. Nevertheless, less than 50% of patients with HFpEF have increased ECV compared to an age-matched population [84]. Despite this consideration, pirfenidone is the first antifibrotic tested in HFpEF patients accounting for a reduction in cardiac fibrosis, with several advantages; it is an oral treatment with a low rate of side effects, and its underlying mechanisms reveal a reduction in profibrotic factors as TGF-β and proinflammatory cytokines, such as TNF-α, IL-4, and IL-13 inhibiting collagen synthesis. Interestingly, in the sub-analysis of the PIROUETTE study, pirfenidone did not reduce inflammatory markers such as GDF-15 [85], revealing a complex interaction between the mechanism underlying inflammation and fibrosis in the pathogenesis of HFpEF. To gain further insight into this potential therapeutic agent’s clinical impact, investigators should carry out a phase 3 trial.

#### 9.2.2. Spironolactone

Another antifibrotic agent (but most known for its neurohormonal effects) widely used in clinical practice and HFrEF is spironolactone. The TOPCAT study first revealed no significant differences in HFpEF patients treated with this agent [86], although later analysis revealed a difference in results when considering the geographical region where patients were recruited [87]. Still, the debate is ongoing about the clinical impact of complex outcomes. In addition, the SPIRIT-HF trial (NCT04727073) is ongoing in which spironolactone is compared with placebo in patients with HFpEF and HFmrEF in a composite of recurrent heart failure hospitalizations and cardiovascular mortality.

#### 9.2.3. Sacubitril/Valsartan

Currently, the ARNICHF trial, studying the role of sacubitril/valsartan as a potential fibrosis modulator, is the only ongoing study aiming for a fibrosis reduction outcome as assessed by CRM in HFpEF (NCT05089539). The results could prompt insights into how this novel drug exerts its protective clinical features. 

#### 9.2.4. Novel Therapies

Other antifibrotic trials are being conducted mainly in the idiopathic pulmonary fibrosis area, such as nintedanib and pamrevlumab; further research is needed for application in HFpEF [88,89]. In pre-clinical models, several molecules are proposed as antifibrotic agents, i.e., salinomycin, an active compound initially used as an anti-coccidial drug, which prevents activation of fibroblasts and promotes cardiomyocyte survival in mice models, but there is a lack of clinical studies to date [90]. Similarly, galectin-3 inhibitors are proposed as potential disease modifiers of HF [91] but no active antagonist compound exists for clinical use. Expectance relays in some studies with novel inhibitors, but still, further research is needed [92].

### 9.3. Anti-Inflammatory Agents

#### 9.3.1. Anti-TNF-α 

As mentioned previously, TNF-α is one of the oldest cytokines described in heart failure, and two classic large trials of anti-TNF-α therapy have been performed in heart failure patients. The RENEWAL [93] (Randomized Etanercept Worldwide Evaluation) trial studied the effects of etanercept, a decoy TNF-α receptor, and the ATTACH [94] (Anti-TNF therapy against congestive heart failure) trial, using infliximab, a neutralizing antibody, with disappointing results not only due to the fact that neither study improved outcomes, but a trend to augmented mortality was observed. Therefore, no studies have been conducted in the specific setting of preserved EF. 

#### 9.3.2. Anti-IL-1

The DHART2 study tested the intervention of direct IL-1 blocker anakinra for 12 weeks aiming for aerobic outcomes but failed to improve peak VO2 and VE/VCO2 slope in a group of obese HFpEF patients [95].

#### 9.3.3. Anti-IL-6

Immune blockade of IL-6 is approved for use in autoimmune diseases with a substantial role in inflammation; a few anti-IL-6 therapies are currently in use for rheumatologic disease, yet there is a lack of studies in heart failure [96].

#### 9.3.4. Colchicine

Colchicine is a potent anti-inflammatory drug used in the treatment of gout and pericarditis, and has properties that relate to the suppression of tubulin polymerization and inflammasome inhibition, thus reducing the production of IL-1β and IL-18. A recent animal model revealed that colchicine alleviates inflammation and improves diastolic dysfunction in heart failure rats with preserved EF model induced by a high-salt diet [97] It is under study for its potential effect on HFpEF patients in the ongoing COLpEF trial (NCT04857931). Its expected results focus on the reduction of inflammatory biomarkers reduction as hs-CRP and, as secondary outcomes, the impact on clinical variables such as NT-proBNP, and troponin, among others. 

#### 9.3.5. iSGLT2

Inhibitors of the sodium–glucose transporter-2 (iSGLT2), such as empagliflozin and dapagliflozin, have prompted special attention and become strongly recommended according to the last guidelines in the treatment of heart failure [28]. Indeed, the promising results in hard outcomes, such as reduction in cardiovascular mortality and hospitalizations in extensive studies in HFrEF [98,99] were someway reproduced in studies in HFpEF [100], lacking the reduction in cardiovascular mortality. Despite the clinical effect, partial knowledge of the underlying mechanism remains elusive [101]. A recent study with LV myocardial samples of HFpEF patients incubated with empagliflozin revealed a diminished expression of adhesion molecules such as I-CAM and V-CAM and lowered release of proinflammatory cytokines such as TNF-a and IL-6, as well as reduced oxidative stress in myocardial fibers [102]. The latter could be a potential mechanism and stands out for inflammation’s central role in HFpEF. Another study revealed similar results in a rat model of HFpEF with sitagliptin, improving inflammation markers such as TNF-a and IL-6 and reducing fibrosis [103].

#### 9.3.6. Metformin

This classical oral treatment for diabetes, tested in many clinical settings, is currently ongoing in a clinical trial evaluating peak oxygen capacity during exercise in HFpEF patients aged older than 60 years (NCT05093959). Inflammatory biomarkers will be measured as a secondary outcome, considering recent results pointing to an anti-inflammatory role for metformin despite the presence of a diabetic status [104].

## 10. Nitric Oxide Pathway and Endothelial Function

### 10.1. NO Pathophysiology and Dysfunction in HF 

Nitric oxide (NO) is a highly reactive molecule that plays a key role in the myocardium by affecting mitochondrial respiration, oxygen consumption, substrate utilization, and myocardial perfusion [105]. It also has a negative inotropic effect in LV, which leads to an improvement in LV distensibility and myocardial performance, leading to earlier relaxation with an increase in its flexibility, as well as its antihypertrophic, antifibrotic, and proangiogenic effects [106,107,108]. In the peripheral vasculature, NO generates vasodilation and mediates anti-proliferative effects; it also improves coronary perfusion and decreases pulmonary vasculature resistance [109].

Levels of NO, and therefore of cGMP, are reduced in HF because of multiple processes [110,111,112]. First, there is decreased L-arginine bioavailability caused by increased arginase activity. Second, there is a downregulation or uncoupling of the NOS3, leading to superoxide formation that contributes to vascular oxidative stress [113]. Third, an alteration of the redox state of sGC is caused by oxidative stress [114].

This pathway has several pharmacological targets that can be divided into nitric oxide donors, sGC modulators (activators and stimulators), phosphodiesterases (PDE) inhibitors, and natriuretic peptide (NP) modulators [113]. 

### 10.2. Novel Trials and Applications Related to the NO-sGC-cGMP Pathway

#### 10.2.1. Nitric Oxide Donors

Despite theoretical and acute benefits described for nitrates, the NEAT-HFpEF trial, isosorbide mononitrate against placebo, led to less physical activity without differences in quality of life [115]. The isosorbide +/− hydralazine group of the MDMA trial failed to prove significant differences against placebo on reflection magnitude, ventricular remodeling, or submaximal exercise [116]. There are other phase II and III trials aiming at the NO pathway that have not published their results yet, including the ONOH trial (NCT02918552), the INABLE trial (NCT02713126), the KNO3CK OUT HFpEF trial (NCT02840799), and the MPMA trial (NCT04913805).

#### 10.2.2. Modulators of sGC

sGC direct stimulators, such as vericiguat and riociguat, act on the NO-sensitive form of sGC and stimulate the enzyme directly by mimicking NO in its absence and sensitizing sGC to low levels of NO [117,118].

The SOCRATES-PRESERVED study tested vericiguat against placebo in HFpEF [119], and despite no significant differences being observed in left atrial volume or NT-proBNP, it was associated with improvements in the quality of life [120,121]. However, further trials (VITALITY-HFpEF) have failed to demonstrate a significant improvement in the physical limitation score [122].

The CAPACITY HFpEF trial proved the effect of praliciguat on functional capacity without significant differences in VO2 peak against placebo [123]. The haemoDYNAMIC trial was another Phase IIb study that assessed the efficacy of riociguat, a novel soluble sGC stimulator that had already been shown to improve exercise capacity and symptoms in patients with pulmonary hypertension (PHT) [124]. Unfortunately, although it enhanced cardiac output, it led to more dropouts than placebo and did not change clinical symptoms in patients with HFpEF [125]. 

#### 10.2.3. PDE Inhibitors

PDE inhibitors reduce pulmonary artery pressure by improving endothelial function and myocardial relaxation [126,127]. PDE-5, which selectively hydrolyses cGMP, is upregulated in HF and cardiac hypertrophy. The use of sildenafil, a PDE-5 inhibitor, had promising results in a small RCT [128], and although some hemodynamic benefit has been observed in the SIDAMI trial [129], it has not shown a significant clinical benefit against placebo in posterior trials [130,131]. Ongoing trials using PDE-5 inhibitors are the ULTIMATE-HFpEF trial (NCT01599117 with idenafil, NCT01726049 with sildenafil), and the PASSION trial (DRKS00014595).

## 11. Mitochondrial and Metabolic Defects 

### 11.1. Some Physiology Aspects

Mitochondria are organelles with a central role in ATP synthesis and cellular metabolism, growth, differentiation, signaling, and death. Mitochondria dynamics involve multiple processes, including fusion and fission, in response to energy demands [132], and producing and managing oxygen-free radicals (ROS). In the myocardium, mitochondria are the primary source of energy, based on fatty acid consumption, and their dysfunction is now regarded as one of the main pathophysiologic mechanisms in the development of HF [133,134,135]. However, peripheral mitochondrial dysfunction in skeletal muscle has been shown to impact myocardial function and play a role in exercise intolerance, a canonical symptom in these patients [39,136]. 

### 11.2. Mitochondrial Dysfunction and Pharmacological Targets

#### 11.2.1. Perhexiline

Perhexiline is a carnitine palmitoyltransferase-1 inhibitor believed to improve cardiac energetics by shifting cardiac metabolism towards the use of carbohydrates, which results in increased mechanical efficiency. There is a lack of results yet (NCT00839228) [137].

#### 11.2.2. Resveratrol

Reduced mitochondrial biogenesis has been described in HF as a consequence of multiple factors. The peroxisome proliferator-activated receptor coactivator (PGC1α) regulates mitochondrial turnover and biogenesis. This pathway is downregulated in HF, which has been suggested as a possible cause of reduced mitochondrial biogenesis [138]. AMPK has been shown to induce mitochondrial biogenesis through direct phosphorylation of PGC1α. Resveratrol, a polyphenol found in red wine that has stimulated mitochondrial biogenesis through AMPK and NO-dependent mechanisms [39], is currently being tested in the GRAPEVINE-HF trial (NCT01185067).

#### 11.2.3. Elamipretide

Cardiolipin is a structural lipid of the mitochondrial internal membrane that anchors to cytochrome c facilitating electron transport between complex III to IV in the electron transport chain (ETC). Cardiolipin is reduced in HF for several reasons [139]. It amplifies the production of ROS by decoupling ETC and generating a vicious cycle that increases the de-peroxidation of cardiolipin [138]. Elamipretide is a mitochondrion-targeting peptide able to act in the inner membrane and act over cardiolipin to restore mitochondrial bioenergetics; it has been shown to normalize mitochondrial function in cell models, as well as to improve LV function in animal models. It is currently being tested in the RESTORE-HF trial (NCT02814097). 

#### 11.2.4. Neladenoson

Neladenoson is a partial adenosine A1 receptor agonist that has shown, in pre-clinical studies, that it improves mitochondrial function and optimizes energy utilization without adverse effects [140]. The PANACHE trial (NCT03098979) could not demonstrate a better dose-related exercise capacity after 20 weeks of analysis.

## 12. Cell Therapy

### 12.1. How It Works?

Cell therapy uses stem cells, progenitor cells, primary cells, or genetically modified cells to replace or repair a damaged or aged tissue or organ. These cells can be administered intravenously, transplanted into the affected region, or recruited from the same tissue to be repaired [141]. In recent decades, it has aroused particular interest, especially in treating several diseases.

In the case of cardiovascular diseases, at the end of the 1990s, the first pre-clinical studies demonstrated the feasibility of transplanting fetal cardiomyocytes [142] and skeletal myoblasts [143] in damaged myocardium and fibrotic scars [144]. Since then, pre-clinical studies have been carried out. Clinicians have started evaluating the potential use of this therapy in ischemic, dilated, and infiltrative cardiomyopathy.

To date, most of the clinical studies that have been carried out to evaluate the efficacy and safety of the use of cell therapy in patients with heart failure have focused mainly on the subgroup of patients with reduced EF [145,146,147,148]. Recently, it was expanded to include a subset of patients with preserved EF.

In the first pre-clinical studies, scientists thought that the mechanism through which cardiac function improved was the replacement of non-functioning cells in the healed myocardium with new myocardium, which achieved electrical integration with the tissue and had contractile capacity [149]. However, later studies suggest that the effect of this therapy was due to indirect action through paracrine interaction with the tissue, either by the release of growth factors or proangiogenic cytokines, resulting in increased collateral circulation and reducing the area of infarction [150,151,152]. This theory is now widely accepted over the idea of direct tissue reconstruction [153].

Different cell types have been studied for their use in cardiovascular diseases, which can be divided into embryonic and adult stem cells. Within this second group, those derived from bone marrow, skeletal muscle, adipose tissue, umbilical cord, and stem cells induced from adult fibroblasts stand out [154]. However, the detailed explanation of each of them exceeds the purposes of this review. Special mention goes to cardiac stem cells and cardiosphere-derived stem cells (spheroids formed from cardiac stem cells in vitro), which have been shown to improve LV function in animal models and preliminary human trials [155,156,157], along with bone marrow-derived hematopoietic stem cells that express CD34+, the results of which have been impressive [154].

### 12.2. Cell Therapy and the Future

Pre-clinical studies in murine HFpEF models induced by hypertension have shown reverse abnormalities, normalization of LV relaxation, and improved survival after a four-week follow-up without reducing hypertension or hypertrophy [158]. In addition, the antiarrhythmic effect has been demonstrated in animal models, with a decrease in atrial arrhythmias, reduced incidence of sudden deaths, and improved survival [159]. Cell therapy is currently being tested in the Regress-HFpEF (NCT02941705) phase 2 trial.

On the other hand, HFpEF patients have a lower level of circulating CD34+ cells [160], which is inversely correlated with diastolic dysfunction [161]. Based on this, Vrtovec et al. conducted a pilot study involving 30 patients with HFpEF. Firstly, all the participants were treated with optimized medical treatment for six months. They then received therapy with CD34+ cells administered transendocardially in a second phase. After a follow-up period of 6 months, they found that CD34+ cell transplantation is feasible and safe and was associated with favorable results based on the measurement of E/e’ ratio, NT-proBNP levels, and the distance walked in the 6 min walk test (6 MWT) [162].

In summary, there are several ongoing trials of new targets from the systemic to the molecular level, as shown in Figure 2. 

## 13. Device-Based Therapies 

### 13.1. Can a Device Ameliorate Impaired Mechanisms in HFpEF? 

The structural alterations determine a small LV when compared to those patients with HFrEF [163] and impaired LV filling function, raising LV end-diastolic pressure, which can be transmitted retrograde to the left atrium (LA), generating atrial remodeling, then to the pulmonary circulation, which produces the symptoms of pulmonary congestion and remodeling [164]. Using devices to treat patients with HFpEF aims to reduce pressure in the LA to normalize hemodynamics would prevent this process. 

The devices used in clinical and pre-clinical studies are interatrial shunt devices (IASDs), which work by forming a shunt between the LA and another cavity to reduce LA pressure and, consequently, retrograde pressure towards the pulmonary circulation [164]; second, LV expander devices, that assist early ventricular filling by optimizing elastic energy during systole and releasing during diastole [165]; third, mechanical circulatory support devices, predominantly the continuous flow rotodynamic blood pumps (RBPs) [166,167] that aim to decompress the left atrium and restore arterial pulsatility and cardiac output; and finally, neurohumoral devices are used to control the autonomic nervous system response in these patients. All these devices and a summary of their ongoing trials are presented in Figure 3.

### 13.2. Interatrial Shunt Devices

In one of the first studies of interatrial shunt devices, the investigators recruited eleven patients with EF greater than 45%, PCWP equal to or greater than 15 mmHg at rest or than 25 mmHg while exercising; and at least one hospitalization for heart failure in the last 12 months, or functional capacity III/IV NYHA dyspnea for at least three months. After 30 days, PCWP decreased by an average of 28%, and the pressure in the right atrium and the systolic pressure of the pulmonary artery did not change. In addition, there was a decrease in RV filling pressure of 5.5 mmHg, and dyspnea improved by two classes in two patients and one class in five patients. In this trial, two serious adverse effects occurred: heart failure requiring re-hospitalization and a malpositioned device [168].

Subsequently, REDUCE LAP-HF, a prospective multicenter phase 1 trial, where 68 patients with similar criteria to the last study were recruited with a 6-month follow-up, shows an improvement in functional capacity (NYHA), better perception of quality of life, and best distance performance in 6 MWT. Structural changes were measured by echocardiography, which shows a decrease in the LV diastolic volume index, an increase in the RV diastolic volume index, an increase in the RA volume, and a slight increase in the LA volume. These structural changes do not affect pro-BNP levels or the adjustment of furosemide therapy [164].

REDUCE LAP-HF I [169], a multicenter phase 2 study and the first to be randomized (to IASD versus a sham procedure) and double-blind, included patients with class III or IV NYHA functional capacity, EF ≥ 40%, PCWP in exercise ≥ 25 mmHg, and PCWP/right atrial pressure gradient ≥ 5 mmHg. Forty-four patients participated, twenty-two for the IASD arm and the control group. In the first month, there was a more significant reduction in PCWP in the intervention group compared to the control group, in all stages of exercise, without a decrease in maximum PCWP. At one year of follow-up [170], a better functional capacity was demonstrated by improved quality of life and the distance walked in 6 MWT. Regarding the structural changes, it was described that at six months, there was already an increase in the RV size in the IASD group versus the control group.

REDUCE LAP-HF II [171] is an ongoing multicenter, randomized, double-blind trial. This study used the same inclusion and exclusion criteria as the previous one, recruiting 608 patients. The primary endpoint was a composite that included cardiovascular mortality or stroke within 12 months, hospitalizations for heart failure or need for intravenous diuretics within 24 months, and changes in the Kansas City Cardiomyopathy Questionnaire from enrollment to 12-month follow-up. Follow-up will be carried out by echocardiography at 6, 12, and 24 months to evaluate the shunt, size, and functionality of cardiac cavities, with a total of 5 years of follow-up. There has yet to be a definitive results publication (NCT03088033).

### 13.3. V-Wave Shunt

V-Wave shunts are devices that allow a unidirectional flow from the LA to the RA when the pressure gradient exceeds 5 mmHg. They are installed percutaneously, similarly to IASDs. The feasibility and operation of the device have been reported in pre-clinical and clinical studies, demonstrating functional and hemodynamic improvements in animal models and humans with HFrEF [172,173].

In a prospective, multicenter study with thirty-eight patients, eight had HFpEF, with a mean follow-up of 28 months, showing an improvement in NYHA functional class up to I and II in 60% of the patients, along with improvements in the quality-of-life questionnaires and 6 MWT. In addition, an adverse event related to the device and three general adverse events with two deaths were described, highlighting a 36% occlusion of the shunts at 12 months of follow-up [174].

### 13.4. Atrial Flow Regulator (AFR)

The AFR is a self-expanding device that allows a variable opening diameter of 6 to 10 mm with an interatrial flow that can be bidirectional, and its installation is percutaneous. The first AFR was implanted in a patient with severe pulmonary hypertension, with improvement in clinical outcomes such as 6 MWT, O_2_ saturation at rest, and symptom relief within six weeks [175].

In the PRELIEVE trial, the safety and effectiveness of AFR were evaluated in HFrEF and HFpEF patients, demonstrating an average decrease of 5 mmHg in PCWP at rest at three months of follow-up [176], this decrease being more significant in patients with HFpEF. Without significant changes in functional capacity, distance walked in 6 MWT, or quality of life, in addition to two major adverse events.

Two studies focused on AFR in patients with HFpEF are currently recruiting: the AFteR registry (NCT04405583) and FROST-HF (NCT03751748).

### 13.5. Trans-Catheter Atrial Shunt Systems

These devices create a left–right interatrial shunt reducing LA pressure. This short-circuit creates an increase in the pressures and volumes of the right cavities. Against this, a device that generates a short circuit from the LA to the coronary sinus is under development without involving the right cavities and avoids the risk of atrial arrhythmias and thromboembolic phenomena.

Initially, eleven patients were recruited to undergo percutaneous atriotomy via the internal jugular vein to generate a shunt from the coronary sinus to the LA. This study included seven patients with HFpEF and four with HFrEF. The 201-day follow-up shows seven patients decreased hospital admissions, improved functional capacity, quality of life, and the 6 WMT. Regarding structural changes, left ventricular function remained unchanged, with a decrease in LA volume, without changes in dimensions, function, or systolic pressure of the right ventricle, and without changes in right atrial pressure, as well as in the mean pressure of the pulmonary artery [177].

Two studies evaluating the Edwards atrial shunt system are currently underway (NCT04965623 and NCT03523416). Participants have already been recruited, and the results will be published early next year.

### 13.6. Left Ventricle Expanders (LVEs)

LVEs are spring-like devices that apply an eccentric force, improving the LV filling capacity. During systole, these devices store elastic energy and transfer it to the LV during diastole, assisting during the early filling phase of the cardiac cycle [178]. There are currently two devices: the ImCardia and the CORolla transapical approach device (CORolla TAA), which were designed to be implanted at the level of the pericardium and endocardium, respectively.

The ImCardia is a self-expanding elastic device that decreases the LV diastolic pressure curve. In addition, animal studies have demonstrated effectiveness in improving filling function [179]. Concerning these findings, a prospective non-randomized study has been developed in nineteen patients with HFpEF, candidates for aortic valve replacement due to aortic stenosis. There were no changes in the LV ejection fraction in the intervention group, but there was a decrease in the LV myocardial mass. The study stopped due to the complexity of installing the device (NCT01347125).

The CORolla TAA stands out over the ImCardia due to its minimally invasive installation through an intercostal incision. In a pre-clinical study, this device resulted in a significant decrease in EF at six months, two mitral regurgitation events, and one rupture of the semi-tendinous chordae. In the histopathological study, the presence of active thrombi was evidenced at 3 and 6 months of follow-up. However, with double platelet anti-aggregation, active thrombi were not documented at follow-ups of 12 and 24 months [179].

The first study with CORolla TAA in HFpEF patients was carried out to evaluate the procedure’s safety, feasibility, and effectiveness in ten patients with a follow-up of 24 months (NCT02499601). Preliminary results in one patient have shown a decrease in LV mass index, left atrial volume, LV end-diastolic volume, functional capacity class, perception of quality of life, and 6 MWT. However, at the 24-month follow-up, these changes tended to decrease.

### 13.7. Mechanical Circulatory Support Devices (MCS)

MCSs provide hemodynamic support to the underperforming left or right ventricles to improve the quality of life of patients with HFrEF in the terminal phase; however, there are no clinical studies to evaluate their efficacy and safety in HFpEF. Examples of these devices are left atrial decompression pumps (Synergy pump, heartware), valveless pulsatile pumps (CoPulse), and the left atrial assist device (LAAD).

#### 13.7.1. Left Atrial Decompression Pumps

Pre-clinical studies have incorporated a pump with the pressure-flow characteristics of the Synergy continuous-flow micropump (HeartWare). Synergy’s blood flow could come from the left atrium or the left ventricle and be ejected directly into the proximal aorta. This study concluded that for HFpEF, this mechanical circulatory support significantly increases cardiac output, provides a modest increase in systolic blood pressure, and markedly reduces left atrial and pulmonary pressures [166].

#### 13.7.2. CoPulse

The valveless pulsatile pump (CoPulse) consists of a device implanted in the apex of the heart that is made up of a blood chamber and an air chamber divided by a flexible polyurethane membrane. Blood flow enters the chamber during diastole and is expelled by displacement of the separating membrane during systole. 

Escher et al. [180] demonstrated in a porcine heart model that this device reduced left atrial pressure and increased cardiac output in vitro.

#### 13.7.3. Left Atrial Assist Device (LAAD)

The LAAD consists of a continuous pump implanted in the mitral valve, transferring blood flow from the left atrium to the left ventricle and was the first MCS device for HFpEF tested in animals. In a pre-clinical study, the LAAD was inserted in four healthy animals through an incision in the left atrium at the level of the mitral valve, resulting in increased cardiac output and mean aortic pressure, with a corresponding decrease in left atrial pressure. In turn, left ventricular end-diastolic pressure, central venous pressure, and heart rate remained stable, and echocardiography did not show left ventricular outflow tract obstruction [181].

### 13.8. Autonomic Regulation

In HFpEF patients, an autonomic imbalance occurs with increased sympathetic activity and decreased vagal control over the heart rate, these changes are responsible for chronic symptoms, such as exercise intolerance, which is a strong determinant of the prognosis and quality of life of these patients [182].

There is significant heterogeneity in sympathetic neural control, such as higher heart rate and sympathetic tone in patients with HFpEF compared to healthy volunteers [183] and increases in muscle sympathetic activity with age in a non-linear way, without significant differences in sex in individuals older than 50 years [182]. In addition, univariate analyses have found a significant correlation between the concentration of arterial norepinephrine with systolic blood pressure and pulmonary capillary pressure, suggesting that the autonomic imbalance could be associated with hemodynamic factors determining the development of this condition [183]. 

Despite the above, different clinical trials focused on this potential therapeutic target, have failed to demonstrate efficacy in this subgroup of patients.

#### 13.8.1. BAROSTIM NEO and BAROSTIM BAT

One of the therapeutic targets with the most potential is a device for baroreceptor reflex activation. The BAROSTIM NEO System is a neuromodulation system that targets the decreased baroreceptor sensitivity observed in HFpEF patients by activating the baroreceptors in the carotid artery wall. This system aims to stimulate the afferent and efferent pathways of the autonomic nervous system, increasing parasympathetic tone and decreasing sympathetic stimuli. This subcutaneous device is implanted by ultrasound-guided incision next to the carotid artery, with an electrode linked to a pulse generator in a subcutaneous pocket, and is secure and effective in HFrEF [184]. Furthermore, the BAROSTIM BAT system has been developed to allow an ultrasound-guided suture-less procedure without cutting down to the carotid artery [185,186].

With a 6-month follow-up in the prospective cohort of the BAROSTIM THERAPY study recruiting HFpEF patients with hypertension resistant to maximally tolerated pharmacological treatment with a diuretic and two other antihypertensive medications, this study will assess changes in systolic blood pressure, as well as LV and LA mass indices, NYHA functional class, and re-hospitalization for HF (NCT02876042).

#### 13.8.2. Cardiac Contractility Modulation (CCM)

This strategy focused on delivering high-voltage non-excitatory biphasic electrical signals in the right ventricular septum during the absolute refractory period of the action potential. Thus, increasing calcium influx into the cardiomyocytes, leading to a sustained increase in contractility without increasing myocardial oxygen consumption. In the long term, it causes changes in gene expression in managing calcium, improving myocardial contractility [187].

Recent studies show a significant benefit in HFpEF patients, improving exercise tolerance and quality of life and decreasing hospitalizations. In light of these results, implantation of the CCM device has recently received FDA approval in HF patients ineligible for CRT, with the Optimizer Smart CE being the only FDA-approved CCM therapy device.

CCM-HFpEF was a prospective multicenter study designed to evaluate the efficacy and safety of CCM therapy in heart failure patients with LVEF ≥ 50% and NYHA class II or III despite 24 weeks of optimal medical therapy. In 47 patients, an improvement was observed in the Kansas City Cardiomyopathy questionnaire and a 67% reduction in the number of hospitalizations for heart failure at one year of follow-up (NCT02895048).

#### 13.8.3. Cardiac Resynchronization Therapy (CRT)

Broadly studied for HFrEF, in a subgroup analysis of the PROSPECT (Predictors of Response to CRT) trial initially designed to test the performance of CRT in patients with HFrEF, revealed similar improvement in patients with LVEF > 35% compared with LVEF < 35% at a 6-month follow-up [188]. This finding promoted other studies, investigating CRT efficacy in patients with LVEF ≥ 40%, such as the PREFECTUS prospective cohort trial that aimed to evaluate CRT effects on chronotropic incompetence in patients with HFpEF (NCT03338374).

Finally, other clinical studies are ongoing, such as the PACE HFpEF trial (NCT04546555). In this trial, the effects of personalized pacing on atrial fibrillation and hospitalization rates will be evaluated, as well as on left atrial and ventricular pressures in patients with HFpEF.

## 14. Conclusions

Several attempts are currently in progress to improve molecular, clinical, and functional outcomes in HFpEF. All of them are based on an in-depth knowledge of pathophysiology of the mechanism of this syndrome because we understood more than twenty years ago when diastolic dysfunction was the hallmark in preserved EF. The goal has been gradually achieved for patients with HFrEF. However, for HFpEF it is not enough. HFpEF is an epidemic disease, similar to HFrEF, but without a therapy that can modify the natural history related to morbidity and mortality. We encourage the working teams to reach new targets that enable the game to be changed for HFpEF patients, even if we must go many times from bench to bedside… and back.

## Figures and Tables

**Figure 1 biomedicines-11-00070-f001:**
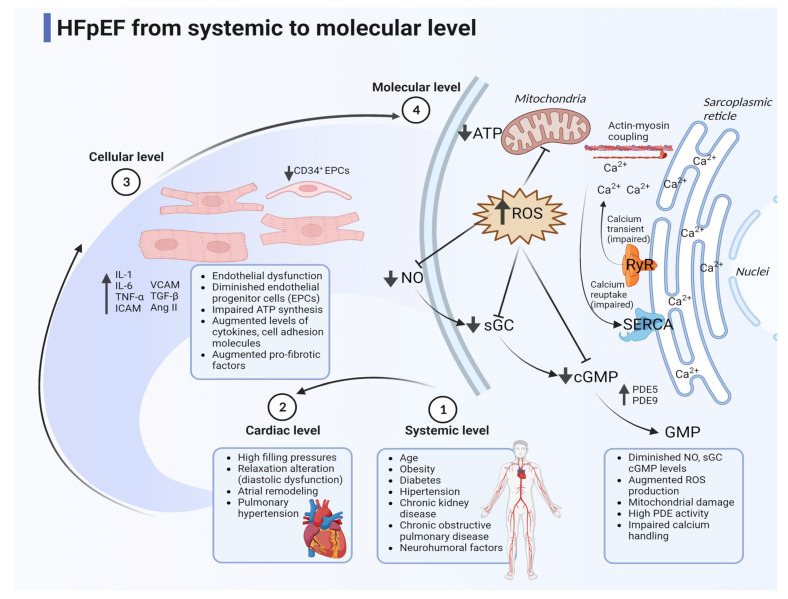
Key mechanisms in HFpEF. Various pathophysiology targets for new therapies use in different pre-clinical and clinical ongoing trials from systemic to the molecular level (IL = interleukin; TNF = tumor necrosis factor; ICAM = intercellular adhesion molecule; VCAM = vascular cell adhesion molecule; TGF = transforming growth factor; AngII = angiotensin II; NO = nitric oxide; sGC = soluble guanylyl cyclase; cGMP = cyclic guanosine monophosphate; ROS = reactive oxygen species; PDE = phosphodiesterase; ATP = adenosine triphosphate; RyR = ryanodine receptor; SERCA = sarco/endoplasmic reticulum Ca^2+^-ATPase) (created with Biorender).

**Figure 2 biomedicines-11-00070-f002:**
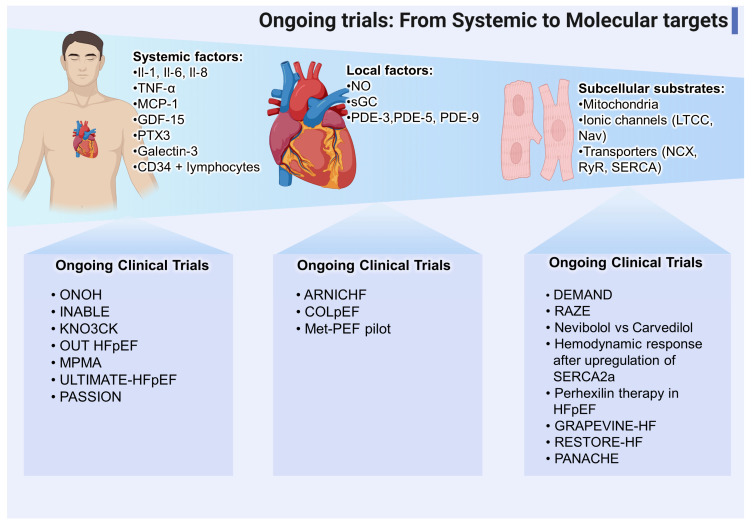
Ongoing clinical trials in HFpEF. Summary of several clinical trials not finished or awaiting results for different levels in the pathophysiology of HFpEF. (IL = interleukin; TNF = tumor necrosis factor; MCP = monocyte chemoattractant protein; GDF = growth differentiation factor; PTX = pentraxine; NO = nitric oxide; sGC = soluble guanylyl cyclase; PDE = phosphodiesterase; LTCC = L-type calcium channel; Nav = voltage-gated sodium channels; NCX = Na^+^-Ca^2+^ exchanger; RyR = ryanodine receptor; SERCA = sarco/endoplasmic reticulum Ca^2+^-ATPase) (created with Biorender).

**Figure 3 biomedicines-11-00070-f003:**
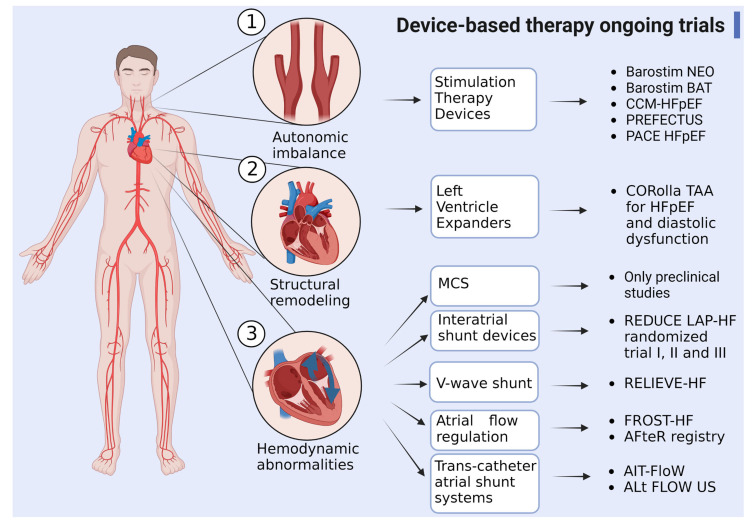
Device-based therapy in HFpEF. Summary of different devices for neurohumoral or mechanical targets in patients with HFpEF and ongoing trials both preclinical and clinical (CCM = cardiac contractility modulation; MCS = mechanical circulatory support) (created with Biorender).

**Table 1 biomedicines-11-00070-t001:** Main Echocardiographic features in patients with HFpEF.

Non-Invasive Complementary Evaluation	Parameters	Cut-Off Values	Comments
Echocardiographic (morphological) [18,19,20]	Left ventricle mass index (g/m^2^) Wall relative thickness Left atrial volume index (mL/m^2^)	Female ≥ 95 Male ≥ 115 >0.42 >34 (sinus rhythm) >40 (atrial fibrillation)	Concentric remodeling and LV hypertrophy support HFpEF diagnosis, but absence of hypertrophy does not exclude it. Left atrial enlargement reflects chronic high filling pressures of LV.
Echocardiographic (functional) [21,22,23,24,25]	E/e’ ratio at rest	>9	The sensitivity and specificity of an E/e’ ratio > 9 was 78% and 59%, compared with 46% and 86% for E/e’ > 13. The mitral E/e’ index correlates with LV stiffness and fibrosis and is less age-dependent than e’.
	E/e’ ratio at peak stress	>15	Exercise echocardiography should be considered abnormal if average E/e’ ratio at peak stress increases to >15, with or without a peak TR velocity > 3.4 m/s.
	TR velocity at rest (m/s)	>2.8	A TR peak velocity > 2.8 m/s indicates increased PASP and is an indirect marker of LV diastolic dysfunction. It has sensitivity 54% and specificity 85% for the presence of HFpEF.
	TR velocity at peak stress (m/s)	>3.4	An increase only in TR velocity in stress should not be used to diagnose HFpEF because it might be caused by a normal hyperdynamic response to exercise (with increased pulmonary blood flow) in the absence of LV diastolic dysfunction.
	PA systolic pressure (mmHg)	>35	PAP > 35 mmHg (derived from tricuspid regurgitation (TR) velocity) was 46% sensitive and 86% specific for HFpEF.
	LV global longitudinal strain	<16%	Reduced LV longitudinal systolic strain and LV early diastolic strain rate have both been identified in HFpEF. The utility of GLS < 16% was moderate (sensitivity 62% and specificity 56%)

LV = left ventricle, E/e’ = early mitral inflow velocity and mitral annular early diastolic velocity ratio, TR = tricuspid regurgitation, PA = pulmonary artery, PAP = pulmonary artery pressure, GLS = global longitudinal strain.

**Table 2 biomedicines-11-00070-t002:** Serology study recommended in HFpEF patients.

Parameters	Cut-Off Values				Comments
BNP (pg/mL) [26]	Acute dyspnea (acute setting)				These are hemodynamic cardiac stress biomarkers. ESC guidelines recommend its use for diagnostic and prognosis in HF.
	<100			HF unlikely	
	100–400			Grey zone	
	>400			HF likely	
	Mild symptoms (chronic setting)				ACC/AHA suggest it for diagnosis, risk stratification (at diagnosis and prior to discharge), and HF prevention.
	<35			HF unlikely	
	35–150			Grey zone	
	>150			HF likely	
NT-proBNP (pg/mL) [27]	Acute dyspnea (acute setting)				Atrial fibrillation, age, acute and chronic kidney disease can reduce its diagnostic accuracy. Lower levels are present in obese patients.
	**<50 years**	**50–75 years**	**>75 years**		
	<300	<300	<300	HF unlikely	
	300–450	300–900	300–1800	Grey zone	
	>450	>900	>1800	HF likely	
	Mild symptoms (chronic setting)				In acute heart failure there is age adjusted cut-off values for NT-proBNP.
	<125			HF unlikely	
	125–600			Grey zone	
	>600			HF likely	
MR-proANP (ng/L) [28]	<120				The BACH and PRIDE trials showed MR-proANP use in the diagnosis of acute HF was similar to BNP and slightly inferior to NT-proBNP, respectively. ESC suggests its use in acute heart failure

BNP = brain natriuretic peptide, NT-proBNP = N-terminal proB-natriuretic peptide, MR-proANP = mid-regional pro-atrial natriuretic peptide. Data was obtained from [26,27,28,29].

**Table 3 biomedicines-11-00070-t003:** Cardiac resonance magnetic imaging findings in different HFpEF etiologies.

Etiology	Main Radiological Characteristics
Ischemic	Subendocardial enhancement in specific coronary artery territory with myocardial fibrosis/scar.
Inflammation (myocarditis, sarcoidosis)	Patchy pattern, late gadolinium enhancement and myocardial oedema. Enhanced T1 and T2 sequences.
Hypertrophic cardiomyopathy	Basal asymmetrical septal hypertrophy (basal anterior septal thickness ≥ 15 mm at end-diastole; ratio of septal to inferolateral wall thickness ≥ 1.3), focal fibrosis on late gadolinium enhancement.
Amyloidosis	Abnormal gadolinium kinetics in the myocardium nulling before the blood pool, enhanced in T1 and ECV.
Hypereosinophilic syndromes	Nonischemic subendocardial scar pattern.

ECV = extracellular volume. This is an original table for this review. Data was obtained from [14,30].

**Table 4 biomedicines-11-00070-t004:** Cut-off for main parameters in heart right catheterization for HFpEF diagnosis.

Parameter	Rest		Exercise ^#^	
	Healthy	HFpEF	Healthy	HFpEF
PCWP (mmHg)	<12	15	<23	≥25
LVEDP (mmHg)	<16	>16	<25	>25
RAP (mmHg)	0–6	>10		>PCWP *
Mean PAP (mmHg)	<20	≥25	<30	>30
PAP/CO slope (mmHg/L/min) ^+^			<3	>3
PCPW/CO slope (mmHg/L/min) ^+^			<2	>2

PCPW = pulmonary capillary wedge pressure, LVEDP = left ventricular end diastolic pressure, RAP = right atrial pressure, PAP = pulmonary artery pressure, CO = cardiac output. ^#^ prognostic value, ^+^ indicators of high risk of cardiovascular death, * RV dysfunction. Data was obtained from [14,31,32,33,34].

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
