# Peer review of "New Opportunities in Heart Failure with Preserved Ejection Fraction: From Bench to Bedside… and Back"

_biomedicines, 2022, doi:10.3390/biomedicines11010070_

Round 1

Reviewer 1 Report

This review describes current pharmacologic and device interventions in patients with HFpEF. The review is very well written, with nice figures.

I only have only one small comment:

- I would not say that spironolactone is best known for its diuretic effects. It is used in patients with HF because of its neurohormonal effects. In addition, the SPIRIT-HF study is ongoing in which spironolactone is compared with placebo in patients with HFpEF and HFmrEF.

Author Response

Dear Sir or Madam,

Thank you for the revision of our article and the comments on our manuscript. We believe that your review helped us to improve the quality of the manuscript and clarify issues that were not appropriately detailed. Accordingly, we are sending a revised version of our manuscript, including the corrections you required. As you will notice, we agree with your comments and concerns, and we are sending you a point-by-point response to your comments.

Thank you for your time and allowing us to resubmit a new version of our manuscript. We look forward to hearing from you and responding to any further questions and comments you may make.

Yours faithfully,

Alfredo Parra-Lucares, MD

Division of Critical Care Medicine

Hospital Clinico Universidad de Chile

Luis Toro, MD PhD FACP

Division of Nephrology

Hospital Clinico Universidad de Chile

Comments and Suggestions for Authors:

This review describes current pharmacologic and device interventions in patients with HFpEF. The review is very well written, with nice figures.

  1. I would not say that spironolactone is best known for its diuretic effects. It is used in patients with HF because of its neurohormonal effects.

We modified in 9.2.2. Spironolactone  “….(but most known for its diuretic effects)…” for “…(but most known for its neurohormonal effects)…”.

  1. In addition, the SPIRIT-HF study is ongoing in which spironolactone is compared with placebo in patients with HFpEF and HFmrEF.

We added at the end of the same paragraph: “In addition, the SPIRIT-HF trial (NCT04727073) is ongoing in which spironolactone is compared with placebo in patients with HFpEF and HFmrEF in a composite of recurrent heart failure hospitalizations and cardiovascular mortality”

Reviewer 2 Report

The subject addressed by the article “New opportunities in heart failure with preserved ejection fraction: from bench to bedside…and back” is of interest for researchers and clinicians. There are a number of reviews on this topic in the last couple of years, some of them being cited by the present one. In the present article authors present both pharmacological treatment and device therapy. Such summary may help to get a general idea, but I think that putting together studies with positive results, negative studies, studies without published results, without a real explanation of the mechanisms make them hard to be followed by the lecturer and impair the value of this review.

Author Response

Reviewer 2

Dear Sir or Madam,

Thank you for the revision of our article and the comments on our manuscript. We believe that your review helped us to improve the quality of the manuscript and clarify issues that were not appropriately detailed. Accordingly, we are sending a revised version of our manuscript, including all the corrections you required. As you will notice, we agree with your comments and concerns, and we are sending you a point-by-point response to your comments.

Thank you for your time and allowing us to resubmit a new version of our manuscript. We look forward to hearing from you and responding to any further questions and comments you may make.

Yours faithfully,

Alfredo Parra-Lucares, MD

Division of Critical Care Medicine

Hospital Clinico Universidad de Chile

Luis Toro, MD PhD FACP

Division of Nephrology

Hospital Clinico Universidad de Chile

Comments and Suggestions for Authors: 

The subject addressed by the article “New opportunities in heart failure with preserved ejection fraction: from bench to bedside…and back” is of interest for researchers and clinicians. There are a number of reviews on this topic in the last couple of years, some of them being cited by the present one. In the present article authors present both pharmacological treatment and device therapy.

  1. “…such summary may help to get a general idea…”

We modified, in according with the editor, the general idea for a comprehensive review in HFpEF with definition, clinical aspects, diagnostic criteria, structural, functional, serological, and hemodynamic features to generate an updated article adding aspects of evidence-based management directed at pathophysiological targets.

  1. “…but I think that putting together studies with positive results, negative studies, studies without published results, without a real explanation of the mechanisms make them hard to be followed by the lecturer and impair the value of this review.”

We reviewed every study in the manuscript and modified some of them. We appreciate your comment to improve the general idea of our work. It is very difficult to make a detailed description for every mechanism by itself (we would like to do this in the future) in just one review, so we preferred detailed all mechanisms instead of describe one by one.

  1. “English very difficult to understand/incomprehensible”

We are aware that English writing is very important for the optimal quality of the review, so the manuscript was undergone extensive English revisions by a native English speaker.
